# Families of Ramanujan-Type Congruences Modulo 4 for the Number of Divisors

**Mircea Merca** 

Department of Mathematical Methods and Models, University  Politehnica of Bucharest, 060042 Bucharest, Romania; mircea.merca@profinfo.edu.ro

**Abstract:** In this paper, we explore Ramanujan-type congruences modulo 4 for the function $\sigma_0(n)$, counting the positive divisors of $n$. We consider relations of the form $\sigma_0\big(8(\alpha n + \beta) + r\big) \equiv 0 \pmod{4}$, with $(\alpha, \beta) \in \mathbb{N}^2$ and $r \in \{1, 3, 5, 7\}$. In this context, some conjectures are made and some Ramanujan-type congruences involving overpartitions are obtained.

**Keywords:** congruences; divisors; overpartitions

**MSC:** 11A25; 11P83



## 1. Introduction

Recall [1] that an overpartition of the positive integer $n$ is an ordinary partition of $n$ where the first occurrence of parts of each size may be overlined. Let $\overline{p}(n)$ denote the number of overpartitions of $n$. For example, the overpartitions of the integer 3 are:

$$3,\ \overline{3},\ 2+1,\ \overline{2}+1,\ 2+\overline{1},\ \overline{2}+\overline{1},\ 1+1+1 \text{ and } \overline{1}+1+1.$$

We see that $\overline{p}(3) = 8$. It is well-known that the generating function of $\overline{p}(n)$ is given by

$$\sum_{n=0}^{\infty} \overline{p}(n) q^n = \frac{(-q;q)_\infty}{(q,q)_\infty}.$$

Here and throughout this paper, we use the following customary $q$-series notation:

$$
(a;q)_n = \begin{cases} 1, & \text{for } n = 0, \\ (1-a)(1-aq)\cdots(1-aq^{n-1}), & \text{for } n > 0; \end{cases}
$$
$$
(a;q)_\infty = \lim_{n\to\infty} (a;q)_n.
$$

Many congruences for the number of overpartitions have been discovered in the recent years by authors such as Chen [2], Chen, Hou, Sun and Zhang [3], Chern and Dastidar [4], Dou and Lin [5], Fortin, Jacob and Mathieu [6], Hirschhorn and Sellers [7], Kim [8,9], Lovejoy and Osburn [10], Mahlburg [11], Xia [12], Xiong [13] and Yao and Xia [14].

Fortin, Jacob and Mathieu [6] founded in 2003 the first Ramanujan-type congruences modulo power of 2 for $\overline{p}(n)$ and for all $n$ that cannot be written as a sum of $s$ or less squares, they obtained that

$$\overline{p}(n) \equiv 0 \pmod{2^{s+1}}. \tag{1}$$

This result is meaningful only for $s < 4$ since, by Lagrange's four-square theorem, all numbers can be written as a sum of four squares. A complete characterization of Ramanujan-type congruences modulo 16 for the overpartition function $\overline{p}(n)$ was provided in 2019 using the function $\sigma_0(n)$ that counts the positive divisors of $n$ [15]. By the proofs of Theorems 1.3 and 1.4 in [15], we easily deduce the following result.

**Theorem 1.** *Let $r \in \{3, 5\}$ be a fixed integer. For all $n \geqslant 0$, we have*

$$\overline{p}(8n + r) \equiv 0 \pmod{16} \quad \Longleftrightarrow \quad \sigma_0(8n + r) \equiv 0 \pmod{4}.$$

In this paper, apart from $\overline{p}(n)$, we consider the overpartition function $\overline{p_o}(n)$ that counts the overpartitions of $n$ into odd parts. The generating function for the number of overpartitions into odd parts is given by

$$\sum_{n=0}^{\infty} \overline{p_o}(n) q^n = \frac{(-q; q^2)_\infty}{(q; q^2)_\infty}. \tag{2}$$

The expression of the generating function for $\overline{p_o}(n)$ was first used by Lebesgue [16] in 1840 in the following series-product identity

$$\sum_{n=0}^{\infty} \frac{(-1; q)_n q^{n(n+1)/2}}{(q; q)_n} = \frac{(-q; q^2)_\infty}{(q; q^2)_\infty}.$$

Although authors such as Bessenrodt [17], Santos and Sills [18] utilized more recently the generating function (2) for $\overline{p_o}(n)$, none of them connected their works to overpartitions into odd parts.

Many congruences for the number of overpartitions into odd parts have been discovered lately [19,20]. It appears that the first Ramanujan-type congruences modulo power of 2 for $\overline{p_o}(n)$ was found in 2006 by Hirschhorn and Sellers [20]. Very recently, Theorem 1 in [21], we introduced a complete characterization of Ramanujan-type congruences modulo 8 for the overpartition function $\overline{p_o}(n)$ considering again the divisor function $\sigma_0(n)$. By the proof of Theorem 1 in [21], we easily deduce the following result.

**Theorem 2.** *Let $r \in \{1, 3\}$ be a fixed integer. For all $n \geqslant 0$, we have*

$$\overline{p_o}(8n + r) \equiv 0 \pmod{8} \quad \Longleftrightarrow \quad \sigma_0(8n + r) \equiv 0 \pmod{4}.$$

Theorems 1 and 2 may be viewed as steps towards classifying all Ramanujan-type congruences for overpartitions, particularly because the divisibility properties of multiplicative functions are more directly accessible with elementary methods than those of functions defined in terms of partitions. Recall that a multiplicative function is an arithmetic function $f(n)$ of a positive integer $n$ with the property that $f(1) = 1$ and $f(ab) = f(a)f(b)$ whenever $a$ and $b$ are coprime.

In this paper, motivated by Theorems 1 and 2, we consider $r \in \{1, 3, 5, 7\}$ to be a fixed integer and investigate pairs $(\alpha, \beta)$ of positive integers for which the following statement is true:

$$\text{For all } n \geqslant 0, \qquad \sigma_0\big(8(\alpha\, n + \beta) + r\big) \equiv 0 \pmod{4}. \tag{3}$$

There is a substantial amount of numerical evidence to conjecture the following.

**Conjecture 1.** *If the statement (3) is true, then there is an odd prime $p$ such that $\alpha$ is divisible by $p^2$ and $8\beta + r$ is divisible by $p$.*

Since a multiplicative function is defined by its values at prime powers, this conjecture boils down to understanding how the divisibility properties of the divisor function $\sigma_0(n)$ at prime powers intersect with arithmetic progressions.

If the statement (3) is true for $(\alpha, \beta)$, then the statement (3) is true for any pair $(k\alpha, b\alpha + \beta)$, with $k \in \mathbb{N}$ and $b \in \{0, 1, \ldots, k - 1\}$. To prove this fact, it is enough to replace $n$ by $kn + b$ in (3). This makes us not very attracted to cases where $\alpha$ is not a square of an odd prime.

**Definition 1.** *For each odd prime p, we define $\mathcal{B}_{r,p}$ to be the set of nonnegative integers $\beta < p^2$ such that*

$$\sigma_0\big(8(p^2 n + \beta) + r\big) \equiv 0 \pmod{4},$$

*for all nonnegative integers n.*

Assuming Conjecture 1, we state the following.

**Conjecture 2.** *For each odd prime p, we have*

$$|\mathcal{B}_{1,p}| = \begin{cases} p - 1, & \text{if } p - 1 \text{ is cubefree,} \\ (p-1)/2, & \text{otherwise.} \end{cases}$$

**Conjecture 3.** *Let $r \in \{3, 5, 7\}$ be a fixed integer. For each odd prime p, we have*

$$|\mathcal{B}_{r,p}| = \begin{cases} (p-1)/2, & \text{if } p \equiv r \pmod{8}, \\ p - 1, & \text{otherwise.} \end{cases}$$

**Conjecture 4.** *Let $r \in \{1, 3, 5, 7\}$ be a fixed integer. Then,*

$$\bigcup_{p \text{ odd prime}} \mathcal{B}_{r,p} = \{n \in \mathbb{N} : \sigma_0(8n + r) \equiv 0 \pmod{4}\} \setminus \begin{cases} \{3\}, & \text{if } r = 3, \\ \varnothing, & \text{otherwise.} \end{cases}$$

Assuming the last conjecture, we remark that there is not an odd prime $p$ such that

$$\sigma_0(8p^2 n + 27) \equiv 0 \pmod{4},$$

for all nonnegative integers $n$.

In this paper, we consider some special cases of our conjectures and present a strategy for proving them. These special cases together with our Theorems 1 and 2 allow us to easily obtain some Ramanujan-type congruences for the overpartition functions $\overline{p}(n)$ and $\overline{p_o}(n)$. Somewhat unrelated to our topics, we will show that these congruences are precursors of stronger congruences. In fact, these stronger congruences were discovered considering few Ramanujan-type congruences modulo 4 for the divisor function $\sigma_0(n)$.

## 2. Some Special Cases

This section is devoted to the presentation of the proof strategy of some special cases of Conjectures 2 and 3 listed bellow. We will rely on the fact that the divisor function $\sigma_0(n)$ is a multiplicative function.

**Theorem 3.**
(i)    $\mathcal{B}_{1,3} = \{4, 7\}$;
(ii)   $\mathcal{B}_{1,5} = \{8, 13, 18, 23\}$.

**Theorem 4.**
(i)    $\mathcal{B}_{3,3} = \{6\}$;
(ii)   $\mathcal{B}_{3,5} = \{4, 14, 19, 24\}$.

**Theorem 5.**
(i)    $\mathcal{B}_{5,3} = \{2, 8\}$;
(ii)   $\mathcal{B}_{5,5} = \{10, 20\}$.

To proof these identities, the following steps have to be performed.
STEP 1. The first step in all our proofs is to verify that for each $\beta \in \mathcal{B}_{r,p}$, $(8\beta + r)/p \in \mathbb{N}$.

STEP 2. For each $\beta \in \mathcal{B}_{r,p}$, we prove that $\gcd(p, 8pn + (8\beta + r)/p) = 1$, for all $n \geqslant 0$.

STEP 3. For each $\beta \in \mathcal{B}_{r,p}$, we prove that $8pn + (8\beta + r)/p$ is not a square, for all $n \geqslant 0$. Thus, for each $\beta \in \mathcal{B}_{r,p}$, we deduce that

$$\sigma_0(8p^2 n + 8\beta + r) = \sigma_0(p)\,\sigma_0\left(8pn + \frac{8\beta + r}{p}\right) \equiv 0 \pmod 4.$$

STEP 4. For each $\beta \in \{0, 1, 2, \ldots, p^2 - 1\} \setminus \mathcal{B}_{r,p}$, we show that there is an integer $n$ such that

$$\sigma_0(8p^2 n + 8\beta + r) \not\equiv 0 \pmod 4.$$

Now, we provide full details for the proofs of Theorems 3–5.

**Proof of Theorem 3.**

(i).

STEP 1. We have $(8 \times 4 + 1)/3 = 11$ and $(8 \times 7 + 1)/3 = 19$.

STEP 2. For all $n \geqslant 0$, it is clear that $\gcd(3, 24n + 11) = 1$ and $\gcd(3, 24n + 19) = 1$.

STEP 3. We suppose that there is an integer $n \geqslant 0$ such that $24n + 11$ is a square. Thus, we deduce that $24n + 11 = (2k + 1)^2$ or $12n + 5 = 2k^2 + 2k$. This identity is not possible, because $12n + 5$ is odd and $2k^2 + 2k$ is even. It is clear that $24n + 11$ cannot be a square. Similarly, it can be proved that $24n + 19$ is not a square. For all $n \geqslant 0$, we deduce that

$$\sigma_0\big(8(9n + 4) + 1\big) = \sigma_0(72n + 33) = \sigma_0(3)\,\sigma_0(24n + 11) \equiv 0 \pmod 4$$

and

$$\sigma_0\big(8(9n + 7) + 1\big) = \sigma_0(72n + 57) = \sigma_0(3)\,\sigma_0(24n + 19) \equiv 0 \pmod 4.$$

STEP 4. Considering that

$$\begin{aligned}
\sigma_0\big(8(9 \times 1 + 0) + 1\big) &\equiv \sigma_0\big(8(9 \times 2 + 1) + 1\big) \equiv \sigma_0\big(8(9 \times 0 + 2) + 1\big) \\
&\equiv \sigma_0\big(8(9 \times 1 + 3) + 1\big) \equiv \sigma_0\big(8(9 \times 0 + 5) + 1\big) \equiv \sigma_0\big(8(9 \times 2 + 6) + 1\big) \\
&\equiv \sigma_0\big(8(9 \times 1 + 8) + 1\big) \equiv 2 \pmod 4,
\end{aligned}$$

the proof is finished.

(ii).

STEP 1. We have $(8 \times 8 + 1)/5 = 13$, $(8 \times 13 + 1)/5 = 21$, $(8 \times 18 + 1)/5 = 29$ and $(8 \times 23 + 1)/5 = 37$.

STEP 2. For all $n \geqslant 0$, it is clear that $\gcd(5, 40n + 13) = 1$, $\gcd(5, 40n + 21) = 1$, $\gcd(5, 40n + 29) = 1$ and $\gcd(5, 40n + 37) = 1$.

STEP 3. We suppose that there is an integer $n \geqslant 0$ such that $40n + 13$ is a square. Thus, we deduce that $40n + 13 = (2k + 1)^2$ or $10n + 3 = k^2 + k$. This identity is not possible, because $10n + 3$ is odd and $k^2 + k$ is even. It is clear that $40n + 13$ cannot be a square. Similarly, it can be proved that $40n + 21$, $40n + 29$ and $40n + 37$ are not squares. For $\beta \in \mathcal{B}_{1,5}$ and $n \geqslant 0$, we deduce that

$$\sigma_0(200n + 8\beta + 1) = \sigma_0(5)\,\sigma_0\left(40n + \frac{8\beta + 1}{5}\right) \equiv 0 \pmod 4.$$

STEP 4. For $\beta \in \{0, 1, \ldots, 24\} \setminus \{\mathcal{B}_{1,5} \cup \{4, 7, 16, 20, 22\}\}$, it is not difficult to check that $\sigma_0\big(8(25 \times 0 + \beta) + 1\big)$ is not congruent to 0 mod 4. In addition, for $\beta \in \{4, 7, 20\}$, we have $\sigma_0\big(8(25 \times 1 + \beta) + 1\big) \not\equiv 0 \pmod 4$. For $\beta \in \{16, 22\}$, we see that $\sigma_0\big(8(25 \times 2 + \beta) + 1\big)$ is not congruent to 0 mod 4. The proof is finished. $\square$

**Proof of Theorem 4.**

(i).

STEP 1. We have $(8 \times 6 + 3)/3 = 17$.

STEP 2. For all $n \geqslant 0$, it is clear that $\gcd(3, 24n + 17) = 1$.

STEP 3. We suppose that there is an integer $n \geqslant 0$ such that $24n + 17$ is a square. Thus, we deduce that $24n + 17 = (2k + 1)^2$ or $3n + 2 = k(k+1)/2$. On the other hand,

$$\frac{k(k+1)}{2} \equiv \begin{cases} 1 \pmod 3, & \text{if } k \equiv 1 \pmod 3 \\ 0 \pmod 3, & \text{otherwise.} \end{cases}$$

It is clear that $24n + 17$ cannot be a square. For all $n \geqslant 0$, we deduce that

$$\sigma_0\big(8(9n + 6) + 3\big) = \sigma_0(72n + 51) = \sigma_0(3)\,\sigma_0(24n + 17) \equiv 0 \pmod 4.$$

STEP 4. Taking into account that

$$\sigma_0\big(8(9 \times 0 + 0) + 3\big) \equiv \sigma_0\big(8(9 \times 0 + 1) + 3\big) \equiv \sigma_0\big(8(9 \times 0 + 2) + 3\big)$$
$$\equiv \sigma_0\big(8(9 \times 1 + 3) + 3\big) \equiv \sigma_0\big(8(9 \times 1 + 4) + 3\big) \equiv \sigma_0\big(8(9 \times 0 + 5) + 3\big)$$
$$\equiv \sigma_0\big(8(9 \times 0 + 7) + 3\big) \equiv \sigma_0\big(8(9 \times 0 + 8) + 3\big) \equiv 2 \pmod 4,$$

the proof is finished.

(ii).

STEP 1. We have $(8 \times 4 + 3)/5 = 7$, $(8 \times 14 + 3)/5 = 23$, $(8 \times 19 + 3)/5 = 31$ and $(8 \times 24 + 4)/5 = 39$.

STEP 2. For all $n \geqslant 0$, it is clear that $\gcd(5, 40n + 7) = 1$, $\gcd(5, 40n + 23) = 1$, $\gcd(5, 40n + 31) = 1$ and $\gcd(5, 40n + 39) = 1$.

STEP 3. We suppose that there is an integer $n \geqslant 0$ such that $40n + 7$ is a square. Thus, we deduce that $40n + 7 = (2k + 1)^2$ or $20n + 3 = 2k^2 + 2k$. This identity is not possible, because $20n + 3$ is odd and $2k^2 + 2k$ is even. It is clear that $20n + 3$ cannot be a square. Similarly, it can be proved that $40n + 23$, $40n + 31$ and $40n + 39$ are not squares. For $\beta \in \mathcal{B}_{3,5}$ and $n \geqslant 0$, we deduce that

$$\sigma_0(200n + 8\beta + 3) = \sigma_0(5)\,\sigma_0\left(40n + \frac{8\beta + 3}{5}\right) \equiv 0 \pmod 4.$$

STEP 4. For $\beta \in \{0, 1, \ldots, 24\} \setminus \big\{\mathcal{B}_{3,5} \cup \{3, 6, 11, 15, 23\}\big\}$, it is not difficult to check that $\sigma_0\big(8(25 \times 0 + \beta) + 3\big)$ is not congruent to 0 mod 4. In addition, for $\beta \in \{3, 6, 23\}$, we have $\sigma_0\big(8(25 \times 1 + \beta) + 3\big) \not\equiv 0 \pmod 4$. For $\beta \in \{11, 15\}$, we see that $\sigma_0\big(8(25 \times 2 + \beta) + 3\big)$ is not congruent to 0 mod 4. The proof is finished. □

**Proof of Theorem 5.**

(i).

STEP 1. We have $(8 \times 2 + 5)/3 = 7$ and $(8 \times 8 + 5)/3 = 23$.

STEP 2. For all $n \geqslant 0$, it is clear that $\gcd(3, 24n + 7) = 1$ and $\gcd(3, 24n + 23) = 1$.

STEP 3. We suppose that there is an integer $n \geqslant 0$ such that $24n + 7$ is a square. Thus, we deduce that $24n + 7 = (2k + 1)^2$ or $12n + 3 = 2k^2 + 2k$. This identity is not possible, because $12n + 3$ is odd and $2k^2 + 2k$ is even. It is clear that $24n + 7$ cannot be a square. Similarly, it can be proved that $24n + 23$ is not a square. For all $n \geqslant 0$, we deduce that

$$\sigma_0\big(8(9n + 2) + 5\big) = \sigma_0(72n + 21) = \sigma_0(3)\,\sigma_0(24n + 7) \equiv 0 \pmod 4$$

and

$$\sigma_0\big(8(9n + 8) + 5\big) = \sigma_0(72n + 69) = \sigma_0(3)\,\sigma_0(24n + 23) \equiv 0 \pmod 4.$$

STEP 4. For $\beta \in \{0, 1, \ldots, 8\} \setminus \mathcal{B}_{5,3}$, it is not difficult to check that $\sigma_0\big(8(9 \times 0 + \beta) + 5\big)$ is congruent to 2 mod 4. The proof is finished.

(ii).

STEP 1. We have $(8 \cdot 10 + 5)/5 = 17$ and $(8 \cdot 20 + 5)/5 = 33$.

STEP 2. For all $n \geqslant 0$, it is clear that $\gcd(5, 40n + 17) = 1$ and $\gcd(5, 40n + 33) = 1$.

STEP 3. We suppose that there is an integer $n \geqslant 0$ such that $40n + 17$ is a square. Thus, we deduce that $40n + 17 = (2k + 1)^2$ or $5n + 2 = k(k + 1)/2$. On the other hand,

$$\frac{k(k+1)}{2} \equiv \begin{cases} 3 \pmod{5}, & \text{if } k \equiv 2 \pmod{5} \\ 1 \pmod{5}, & \text{if } k \equiv \{1, 3\} \pmod{5} \\ 0 \pmod{5}, & \text{otherwise.} \end{cases}$$

It is clear that $40n + 17$ cannot be a square. Similarly, we suppose that there is an integer $n \geqslant 0$ such that $40n + 33$ is a square. Thus, we deduce that $40n + 33 = (2k + 1)^2$ or $5n + 4 = k(k + 1)/2$. Because $k(k + 1)/2 \not\equiv 4 \mod 5$, this identity is not possible. For $\beta \in \mathcal{B}_{5,5}$ and $n \geqslant 0$, we deduce that

$$\sigma_0(200n + 8\beta + 5) = \sigma_0(5)\, \sigma_0\left(40n + \frac{8\beta + 5}{5}\right) \equiv 0 \pmod{4}.$$

STEP 4. For $\beta \in \{0, 1, \ldots, 24\} \setminus \{\mathcal{B}_{5,5} \cup \{2, 8, 9, 11, 15, 16, 17, 23\}\}$, it is not difficult to check that $\sigma_0(8(25 \times 0 + \beta) + 5)$ is congruent to 2 mod 4. In addition, for $\beta \in \{8, 9, 11, 15, 16, 23\}$, we have $\sigma_0(8(25 \times 1 + \beta) + 5) \equiv 2 \pmod{4}$. For $\beta \in \{2, 17\}$, we see that $\sigma_0(8(25 \times 2 + \beta) + 5)$ is congruent to 2 mod 4. The proof is finished. $\square$

It seems that the approach outlined in Steps 1, 2 and 4 can be easily automated. Unfortunately, we cannot say the same about Step 3 because we do not have a criterion which establishes the parity of $(8\beta + r)/p$. Is the number $(8\beta + r)/p$ always odd? When $(8\beta + r)/p$ is an odd number, we need to investigate identities of the form

$$8pn + \frac{8\beta + r}{p} - 1 = 4k(k + 1).$$

When $(8\beta + r)/p$ is an even number, we need to investigate identities of the form

$$8pn + \frac{8\beta + r}{p} = 4k^2.$$

Can the investigation of these identities be automated? We do not have an answer to this question yet.

## 3. Some Ramanujan-Type Congruences

Let $a(n)$ be a sequence of integers defined by

$$\sum_{n=0}^{\infty} a(n)\, q^n = \prod_{\delta | M} (q^\delta; q^\delta)_\infty^{r_\delta}, \tag{4}$$

where $M$ is a positive integer and $r_\delta$ are integers. Based on the ideas of Rademacher [22], Newman [23,24] and Kolberg [25], Radu [26] developed in 2009 an algorithm to verify the congruences

$$a(mn + t) \equiv 0 \pmod{u},$$

for any given $m$, $t$ and $u$, and for all $n \geqslant 0$.

In 2015, Radu [27] constructed an algorithm, called the Ramanujan–Kolberg algorithm, to derive identities on the generating functions of $a(mn + t)$ using modular functions for $\Gamma_0(N)$. A description of the Ramanujan–Kolberg algorithm can be found in Paule and Radu [28]. Recently, Smoot [29] provided a successful Mathematica implementation of Radu's algorithm. This package is called `RaduRK`.

In this section, we use the `RaduRK` package to obtain some Ramanujan-type congruences for the overpartition functions $\overline{p}(n)$ and $\overline{p_o}(n)$. According to Theorems 2 and 3, we can write the following result.

**Corollary 1.** *For* $n \equiv \{4, 7\}$ (mod 9) *or* $n \equiv \{8, 13, 18, 23\}$ (mod 25), *we have*

$$\overline{p_o}(8n + 1) \equiv 0 \pmod{8}.$$

Upon reflection, one expects that there might be a stronger result.

**Theorem 6.**

*(i)* *For all* $n \equiv \{4, 7\}$ (mod 9), *we have*

$$\overline{p_o}(8n + 1) \equiv 0 \pmod{24}.$$

*(ii)* *For all* $n \equiv \{8, 13, 18, 23\}$ (mod 25), *we have*

$$\overline{p_o}(8n + 1) \equiv 0 \pmod{32}.$$

**Proof.** The generating function for $\overline{p_o}(n)$ can be written as

$$\frac{(q^2; q^2)_\infty^3}{(q; q)_\infty^2 \, (q^4; q^4)_\infty}.$$

This can be described by setting $M = 4$ and $r_1 = -2$, $r_2 = 3$, $r_4 = -1$.

(i) Considering the `RaduRK` program with

```
RK[12,4,{-2,3,-1},72,33]
```

and

```
RK[12,4,{-2,3,-1},72,57],
```

we deduce that

$$\sum_{n=0}^{\infty} \overline{p_o}(72n + 33) \, q^n \equiv 0 \pmod{24}$$

and

$$\sum_{n=0}^{\infty} \overline{p_o}(72n + 57) \, q^n \equiv 0 \pmod{24}.$$

(ii) To obtain the second congruence identity, we consider the `RaduRK` program with

```
RK[2,4,{-2,3,-1},200,65]
```

and

```
RK[2,4,{-2,3,-1},200,105].
```

We deduce that

$$\left( \sum_{n=0}^{\infty} \overline{p_o}(200n + 65) \, q^n \right) \left( \sum_{n=0}^{\infty} \overline{p_o}(200n + 185) \, q^n \right) \equiv 0 \pmod{2^{10}}$$

and

$$\left( \sum_{n=0}^{\infty} \overline{p_o}(200n + 105) \, q^n \right) \left( \sum_{n=0}^{\infty} \overline{p_o}(200n + 145) \, q^n \right) \equiv 0 \pmod{2^{10}}.$$

Having

$$\overline{p_o}(65) = 2^5 \times 16\,851,$$
$$\overline{p_o}(200 + 105) = 2^5 \times 6\,293\,025\,198\,351,$$
$$\overline{p_o}(145) = 2^5 \times 64\,201\,703,$$
$$\overline{p_o}(185) = 2^5 \times 1\,713\,260\,289,$$

for $\alpha \in \{65, 105, 145, 185\}$, we notice that

$$\sum_{n=0}^{\infty} \overline{p_o}(200n + \alpha) \, q^n \not\equiv 0 \quad (\mathrm{mod}\ 2^6)$$

and

$$\sum_{n=0}^{\infty} \overline{p_o}(200n + \alpha) \, q^n \equiv 0 \quad (\mathrm{mod}\ 2^5).$$

This concludes the proof. □

According to Theorems 1, 2 and 4, we can write the following result.

**Corollary 2.** *For $n \equiv 6 \pmod 9$ or $n \equiv \{4, 14, 19, 24\} \pmod{25}$, we have*

$$\overline{p}(8n + 3) \equiv 0 \pmod{16} \qquad and \qquad \overline{p_o}(8n + 3) \equiv 0 \pmod 8.$$

There are stronger results.

**Theorem 7.**

*(i)    For all $n \equiv 6 \pmod 9$, we have*

$$\overline{p_o}(8n + 3) \equiv 0 \pmod{24}.$$

*(ii)    For all $n \equiv \{4, 14, 19, 24\} \pmod{25}$, we have*

$$\overline{p_o}(8n + 3) \equiv 0 \pmod{64}.$$

**Proof.** (i) To obtain the first congruence identity, we consider the RaduRK program with

```
RK[4,4,{-2,3,-1},72,51]
```

and obtain

$$\sum_{n=0}^{\infty} \overline{p_o}(72n + 51) \, q^n \equiv 0 \pmod{24}.$$

(ii) To obtain the second congruence identity, we consider again the RaduRK program with

```
RK[2,4,{-2,3,-1},200,35]
```

and

```
RK[2,4,{-2,3,-1},200,155].
```

These give us

$$\left( \sum_{n=0}^{\infty} \overline{p_o}(200n + 35) \, q^n \right) \left( \sum_{n=0}^{\infty} \overline{p_o}(200n + 115) \, q^n \right) \equiv 0 \pmod{2^{12}}$$

and

$$\left(\sum_{n=0}^{\infty} \overline{p_o}(200n + 155)\, q^n\right)\left(\sum_{n=0}^{\infty} \overline{p_o}(200n + 195)\, q^n\right) \equiv 0 \quad (\mathrm{mod}\ 2^{12}).$$

Having

$$\overline{p_o}(35) = 2^6 \times 113,$$
$$\overline{p_o}(115) = 2^6 \times 2\,041\,219,$$
$$\overline{p_o}(200 + 155) = 2^6 \times 59\,890\,735\,496\,633,$$
$$\overline{p_o}(195) = 2^6 \times 1\,844\,065\,971,$$

for $\alpha \in \{35, 115, 155, 195\}$, we notice that

$$\sum_{n=0}^{\infty} \overline{p_o}(200n + \alpha)\, q^n \not\equiv 0 \quad (\mathrm{mod}\ 2^7)$$

and

$$\sum_{n=0}^{\infty} \overline{p_o}(200n + \alpha)\, q^n \equiv 0 \quad (\mathrm{mod}\ 2^6).$$

This concludes the proof. $\square$

**Theorem 8.** *For all* $n \equiv \{19, 24\}$ (mod 25), *we have*

$$\overline{p}(8n + 3) \equiv 0 \quad (\mathrm{mod}\ 160).$$

**Proof.** To obtain this congruence identity, we consider the `RaduRK` program with

`RK[2,2,{-2,1},200,155].`

This gives us

$$\left(\sum_{n=0}^{\infty} \overline{p}(200n + 155)\, q^n\right)\left(\sum_{n=0}^{\infty} \overline{p}(200n + 195)\, q^n\right) \equiv 0 \quad (\mathrm{mod}\ 25600).$$

Having

$$25600 = 2^{10} \times 5^2,$$
$$\overline{p}(155) = 2^5 \times 5 \times 3^2 \times 13 \times 1693 \times 2\,402\,791,$$
$$\overline{p}(195) = 2^5 \times 5 \times 3 \times 6091 \times 2\,417\,744\,023,$$

for $\alpha \in \{155, 195\}$, we notice that

$$\sum_{n=0}^{\infty} \overline{p}(200n + \alpha)\, q^n \not\equiv 0 \quad (\mathrm{mod}\ 2^6)$$

and

$$\sum_{n=0}^{\infty} \overline{p}(200n + \alpha)\, q^n \not\equiv 0 \quad (\mathrm{mod}\ 5^2).$$

Thus, for $\alpha \in \{155, 195\}$, we deduce that

$$\sum_{n=0}^{\infty} \overline{p}(200n + \alpha)\, q^n \equiv 0 \quad (\mathrm{mod}\ 2^5 \cdot 5).$$

This concludes the proof. □

According to Theorems 1 and 5, we can write the following result.

**Corollary 3.** *For $n \equiv \{2, 8\} \pmod 9$ or $n \equiv \{10, 20\} \pmod{25}$, we have*

$$\overline{p}(8n + 5) \equiv 0 \pmod{16}.$$

There are stronger results.

**Theorem 9.** *For all $n \equiv 8 \pmod 9$, we have*

$$\overline{p}(8n + 5) \equiv 0 \pmod{32}.$$

**Proof.** To obtain this congruence identity, we consider the `RaduRK` program with

```
RK[2,2,{-2,1},72,69].
```

This gives us

$$\sum_{n=0}^{\infty} \overline{p}(72n + 69)\, q^n \equiv 0 \pmod{32}.$$

□

## 4. Open Problems and Concluding Remarks

In this paper, we show that each odd prime generates four families of Ramanujan-type congruences modulo 4 for the number of divisors. Assuming Conjecture 1, the algorithm for generating $\mathcal{B}_{r,p}$ is not difficult because $8\beta + r$ must be a multiple of the odd prime $p$. Related to the case $r = 1$ of Conjecture 4, we remark that there is a substantial amount of numerical evidence to conjecture the following.

**Conjecture 5.** *If $n$ is an integer that is not the difference between a triangular number and a square number, then*

$$\sigma_0(8n + 1) \equiv 0 \pmod 4.$$

We focused on the cases $(\alpha, \beta)$, where $\alpha$ is the square of an odd prime. When $\alpha$ is a multiple of the square of an odd prime, we can derive other pairs $(\alpha', \beta')$ for which the statement (3) is true. For example, considering $\mathcal{B}_{1,3} = \{4, 7\}$, we easily deduce that the statement (3) is true if

$$\begin{aligned}
(\alpha, \beta) \in \{ & (81, 4), (81, 7), (81, 13), (81, 16), (81, 22), (81, 25), \\
& (81, 31), (81, 34), (81, 40), (81, 43), (81, 49), (81, 52), \\
& (81, 58), (81, 61), (81, 67), (81, 70), (81, 76), (81, 79) \}.
\end{aligned}$$

We remark that there are two pairs, $(81, 37)$ and $(81, 64)$, which cannot be derived from the pairs $(9, 4)$ or $(9, 7)$. In addition, we remark that

$$\sigma_0\big(8(81n + 37) + 1\big) = \sigma_0\big(27(24n + 11)\big) \equiv 0 \pmod 8$$

and

$$\sigma_0\big(8(81n + 64) + 1\big) = \sigma_0\big(27(24n + 19)\big) \equiv 0 \pmod 8,$$

for all $n \geqslant 0$. The proof of these congruences follows easily if we consider that

$$\gcd(27, 24n + 11) \qquad \text{and} \qquad \gcd(27, 24n + 19) = 1,$$

for all $n \geqslant 0$. Moreover, $24n + 11$ and $24n + 19$ cannot be squares.

The study of congruences of the form

$$\sigma_0(8n + r) \equiv 0 \pmod{2^k},$$

with $r \in \{1, 3, 5, 7\}$, can be a very appealing topic. In analogy with (3), we can consider the following statement:

$$\text{For all } n \geqslant 0, \qquad \sigma_0\big(8(\alpha\, n + \beta) + r\big) \equiv 0 \pmod{2^k}. \tag{5}$$

There is a substantial amount of numerical evidence to state the following generalization of Conjecture 1.

**Conjecture 6.** *If the statement (5) is true, then there is a sequence of odd prime numbers, $p_1 \leqslant p_2 \leqslant \ldots \leqslant p_{k-1}$, such that $\alpha$ is divisible by $(p_1 p_2 \cdots p_{k-1})^2$ and $8\beta + r$ is divisible by $p_1 p_2 \cdots p_{k-1}$.*

On the other hand, our investigations indicate that Conjecture 6 can be generalized if we consider congruences of the form

$$\sigma_0(\alpha n + \beta) \equiv 0 \pmod{2^k}.$$

In analogy with (5), we can consider the following statement:

$$\text{For all } n \geqslant 0, \qquad \sigma_0(\alpha\, n + \beta) \equiv 0 \pmod{2^k}. \tag{6}$$

We state the following generalization of Conjecture 6.

**Conjecture 7.** *If the statement (6) is true, then there is a sequence of prime numbers, $p_1 \leqslant p_2 \leqslant \ldots \leqslant p_{k-1}$, such that $\alpha$ is divisible by $(p_1 p_2 \cdots p_{k-1})^2$ and $\beta$ is divisible by $p_1 p_2 \cdots p_{k-1}$.*

Because $\sigma_0(n)$ is a multiplicative function, these conjectures motivate the question of identifying all Ramanujan-type congruences for multiplicative functions.

**Funding:** This research received no external funding.

**Institutional Review Board Statement:** Not applicable.

**Informed Consent Statement:** Not applicable.

**Data Availability Statement:** Not applicable.

**Conflicts of Interest:** The author declares no conflict of interest.

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
