# Peer review of "Families of Ramanujan-Type Congruences Modulo 4 for the Number of Divisors"

_axioms, doi:10.3390/axioms11070342_

Round 1
Reviewer 1 Report
Review report on “Families of Ramanujan-type congruences modulo 4 for the
number of divisors”
In the manuscript, the author studies Ramanujan-type congruences modulo 4 for the divisor function. Based on the previous results by the author, these congruences are related to Ramanujan-type congruences modulo 16 for the overpartition functions $\bar{p}(n)$ and congruences modulo 8 for the overpartitions into odd parts $\bar{p}_o(n)$.
In the introduction, the author clearly motivated the research question, i.e., Conjecture 1 and its followers, and provided relevant references for background. The key results of the paper is to prove some special cases of the conjecture, and its implications for the overpartition functions. In Section 3, the author utilizes Radu’s algorithm RaduRK to derive some congruences for overpartition functions. Section 4 provides open problems and some remarks.
In my opinion, Ramanujan-type congruences for the overpartition functions is always an interesting research question in number theory. The paper provides some new perspectives on the relevant overpartition functions via the divisor function, especially Conjecture 1. Unfortunately, I have some concerns regarding the method of the paper.
First, the current introduction does not provide enough details for the method used by the author. In particular, what is the strength and weakness of the method in terms of proving the conjecture?
Second, Section 2 only provides parts of the proofs for Theorem 3-5. I suggest the author to make clear whether or not the remainder of the proofs follow from the same strategy. It seems that the approach outlined in Section 2 can be automated and the conjecture can be verified accordingly. The author might want to discuss how his method works for larger r and p.
Third, Section 3 utilizes Radu’s algorithm to prove some congruences for overpartition functions. However it is unclear from the writeup that how these stronger congruences are helpful or related to the topic of the paper, congruences for the divisor function.
All in all, I recommend the paper to be revised. I include more minor and precise suggestions for changes below.
Page 3 “we remark that is no and odd prime p such that” the sentence reads odd.
Page 3 line 51 “B_{p,r,}” mistyped, should be “B_{r,p}”, the similar mistypes happen throughout the manuscript.
All in all, I recommend the paper to be revised.
Author Response
Reviewer. First, the current introduction does not provide enough details for the method used by the author.
Response. Theorems \ref{L1} and \ref{T1} may be viewed as steps towards classifying all Ramanujan-type congruences for overpartitions, particularly because the divisibility properties of multiplicative functions are more directly accessible with elementary methods than those of functions defined in terms of partitions. Recall that a multiplicative function is an arithmetic function $f(n)$ of a positive integer $n$ with the property that $f(1)$ = 1 and $f(ab)=f(a)f(b)$ whenever $a$ and $b$ are coprime.
Reviewer. In particular, what is the strength and weakness of the method in terms of proving the conjecture?
Response. Since a multiplicative function is defined by its values at prime powers, this conjecture boils down to understanding how the divisibility properties of the divisor function $\sigma_0(n)$ at prime powers intersect with arithmetic progressions.
Reviewer. Second, Section 2 only provides parts of the proofs for Theorem 3-5. I suggest the author to make clear whether or not the remainder of the proofs follow from the same strategy. It seems that the approach outlined in Section 2 can be automated and the conjecture can be verified accordingly. The author might want to discuss how his method works for larger r and p.
Response. We provide the complete proofs for Theorem 3-5. It seems that the approach outlined in Steps 1,2 and 4 can be easily automated. Unfortunately, we cannot say the same about Step 3 because we do not have a criterion which establishes the parity of $(8\beta+r)/p$. Is the number $(8\beta + r) / p$ always odd? When $(8\beta+r)/p$ is an odd number, we need to investigate identities of the form $$8pn+\frac{8\beta+r}{p}-1 =4k(k+1).$$ When $(8\beta+r)/p$ is an even number, we need to investigate identities of the form
$$8pn+\frac{8\beta+r}{p} =4k^2.$$ Can the investigation of these identities be automated? We do not have an answer to this question yet.
Reviewer. Third, Section 3 utilizes Radu’s algorithm to prove some congruences for overpartition functions. However it is unclear from the writeup that how these stronger congruences are helpful or related to the topic of the paper, congruences for the divisor function.
Response. In this paper, we consider some special cases of our conjectures and present a strategy for proving them. These special cases together with our Theorems \ref{L1} and \ref{T1} allow us to easily obtain some Ramanujan-type congruences for the overpartition functions $\overline{p}(n)$ and $\overline{p_o}(n)$. Somewhat unrelated to our topics, we will show that these congruences are precursors of stronger congruences. In fact, these stronger congruences were discovered considering few Ramanujan-type congruences modulo $4$ for the divisor function $\sigma_0(n)$.

Reviewer 2 Report
Author should wrote the specific motivation of the paper.
Give the proof of Theorem 3(ii), Theorem 4(ii) and Theorem 5(ii).
Author should provide at least one supportive example in regard to Theorem 3, Theorem 4 and Theorem 5 for the both (i) and (ii).
Can you formulate the Theorem 3, Theorem 4 and Theorem 5 in terms of overpartition function?
Write the section of conclusion.
Author Response
Reviewer. Author should wrote the specific motivation of the paper.
Response. Theorems \ref{L1} and \ref{T1} may be viewed as steps towards classifying all Ramanujan-type congruences for overpartitions, particularly because the divisibility properties of multiplicative functions are more directly accessible with elementary methods than those of functions defined in terms of partitions. Recall that a multiplicative function is an arithmetic function $f(n)$ of a positive integer $n$ with the property that $f(1)$ = 1 and $f(ab)=f(a)f(b)$
whenever $a$ and $b$ are coprime.
Reviewer. Give the proof of Theorem 3(ii), Theorem 4(ii) and Theorem 5(ii). Author should provide at least one supportive example in regard to Theorem 3, Theorem 4 and Theorem 5 for the both (i) and (ii).
Response. We provide the complete proofs for Theorems 3-5.
Reviewer. Can you formulate the Theorem 3, Theorem 4 and Theorem 5 in terms of overpartition function?
Response. See Corollaries 1-3.
Reviewer. Write the section of conclusion.
Response. We have update the section "Open problems and concluding remarks"

Round 2
Reviewer 1 Report
Most of my previous concerns are addressed and/or clarified properly. I find the revisions adequate and have no further comments.
Reviewer 2 Report
Recommend in the present form.